# Sedentary Behavior and Pain after Physical Activity in Women with Fibromyalgia—The Influence of Pain-Avoidance Goals and Catastrophizing

**DOI:** 10.3390/biomedicines11010154

**Published:** 2023-01-07

**Authors:** Lorena Gutiérrez, Carmen Écija, Patricia Catalá, Cecilia Peñacoba

**Affiliations:** Department of Psychology, Universidad Rey Juan Carlos, Avenida de Atenas S/N, 28922 Alcorcón, Spain

**Keywords:** fibromyalgia, sedentarism, conflict goals, pain catastrophizing, motivational interventions, moderated-mediation model

## Abstract

Background: Fibromyalgia is characterized by chronic pain and fatigue that triggers a functional disability caused by the lack of activity. Pain catastrophizing may contribute to avoiding activity with the intention of managing pain levels. Based on the sedentary behavior with fibromyalgia, the present study assessed the preference of pain-avoidance goals and pain catastrophizing as mediator and moderator variables, respectively, that influence pain perception after a 6-min-walking test. Methods: The sample was composed of 76 women with fibromyalgia (mean age = 55.05, SD = 7.70). Previous sedentary behavior, preference for pain-avoidance goals, and pain catastrophizing were evaluated before starting the walking-test. Subsequently, pain perception was evaluated. Results: A significant moderated-mediation model was found in which pain-avoidance goals mediated the relationship between sedentarism and pain after a walking-test, and pain catastrophizing moderated the relationship between the preference for pain-avoidance goals and pain perception. Specifically, high levels of pain catastrophizing contributed to increased pain perceptions after completing the test (B = 0.570, *p* = 0.03, CI 95% (0.09, 0.11)]. Conclusions: The results suggest that motivational interventions can improve the symptoms because their objectives are focused on managing conflict goals. These interventions should focus on catastrophic cognitions considering that pain catastrophizing is deemed to be one of the major inhibitors of physical activity in fibromyalgia.

## 1. Introduction

Fibromyalgia (FM) is a physical condition characterized by chronic widespread pain, often associated with poor sleep, fatigue, depression, reduction of physical functionality, and with a multitude of personal costs that generate suffering in patients [1,2]. Among these costs are the affective and cognitive problems that accompany the patients’ lack of knowledge of their disease due to the unclear etiology [3]. Approximately 4% of all chronic pain problems are usually diagnosed as fibromyalgia, of which 90% of these diagnoses are in the female population [1]. This health problem seems to be more frequent in women between 40 and 60 years of age [4,5]. Given its complexity due to the presence of comorbid problems such as anxiety and/or depression, it makes the clinical approach to improve the quality of life of those affected with fibromyalgia particularly difficult. Therefore, one of the main objectives that have been addressed by alternative treatments has been the control and management of pain with the intention of improving physical functionality and the quality of life [6].

However, focusing the therapeutic efficacy on reducing pain severity through inactivity is an insufficient long-term goal [7]. Chronic pain impairs the ability to achieve the goals that people value in their life. Thus, pharmacological interventions have shown to be effective in the short term, but these are insufficient for improving the patients’ quality of life in the long term [4,8]. In this context, the role of physical activity and exercise have shown positive effects on the physical and psychological health of chronic pain patients [9,10]. Therefore, the therapeutic goal should focus further on increasing physical activity rather than the promotion of inactivity.

Due to the complexity of the disease, it is important to develop and implement multidisciplinary treatments that are methodologically rigorous, focused on reducing the impact of FM [11,12]. Despite the difficulty of agreeing on the most effective treatment [13], it seems that the biopsychosocial approach that includes educational, cognitive–behavioral, physical exercise, and medication guidelines is the one most supported by clinical evidence [14,15]. Multicomponent programs that include cognitive–behavioral therapies and physical exercise have been shown to affect the perceived impact after 12 months [14,16]. One of the reasons why the results are not maintained in the long term is due to the lack of incorporation of medication. However, in current studies that have designed multicomponent programs that include pharmacological therapy, they also fail to identify the mechanism of action that facilitates positive results in patients in the long term. Therefore, long-term benefits are observed in patients receiving physical activity-based therapy, which supports the importance of promoting physical exercise adapted to chronic pain [15].

Previous studies suggested that women with FM are less active compared to people without previous pathologies [17,18]. Preliminary relationships between physical activity and pain modulation suggest that sedentary behaviors (e.g., sitting for hours in a day) may contribute to the onset, maintenance, and severity of chronic pain. As far as we know, the lack of adherence of FM patients to physical activity lies in the lack of adaptation of the intensity of physical activity to the specific conditions of each of these patients [19]. For this reason, walking behavior is warranted as an ideal form of aerobic activity owing to its ease of having a relatively low impact. It has a low risk of musculoskeletal injury and it is recommended for previously sedentary people [20]. In addition, walking regularly significantly reduces pain, fatigue, depression, and functional impairment compared to pharmacological treatments [21,22,23].

Even though walking behavior is a pattern adapted to the condition of FM patients, physical dysfunction continues to be a main problem preventing an improved quality of life of these patients [24]. Traditional models of the development of pain explain that people’s interpretations of pain lead to excessive fear, the result of which is the avoidance of any physical activity, increased pain intensity, and physical disability [25]. Concurrent with this, pain catastrophizing, referring to the tendency to exaggerate the threat value of pain and its consequences, has been widely studied [26].

In neurobiological terms, catastrophic beliefs about pain act as anticipatory to the painful sensation. This translates into the rapidity of the dorsal horn activation, so that painful information is transmitted in an accelerated manner in top-down processes [27,28,29]. Pain catastrophizing is associated with self-reports of pain consequences such as disability from migraines, rheumatic diseases, low back pain, and fibromyalgia. Thus, it is suggested that pain catastrophizing has prognostic value in the maintenance and development of chronic pain. In addition to the neurobiological explanation of pain catastrophizing as a mechanism that influences pain perception, there are neural correlates that provide evidence of brain activity related to pain and pain catastrophizing. A recent meta-analysis, which compiled neuroimaging studies, analyzed the mechanisms underlying pain catastrophizing [30], and found a significant association of perceived/self-reported pain with the brain areas involved in pain perception: areas related to sensory information (the primary and secondary somatosensory cortex, posterior insula, and hypothalamus), as well as affective/emotional brain areas (the anterior cingulate cortex and anterior insula) [31,32].

However, the link between pain catastrophizing and pain is more complex because other contextual variables significantly influence this relationship [33]. In the reviewed literature, the incorporation of the behavior of walking is problematic due to pain catastrophizing and the influence of other motivational variables, such as fear and avoidance, as multiple personal goals occur in a motivational context and, on many occasions, arise in competition [34,35]. Affective-motivational theories about pain understand that FM patients manage different goals that are incompatible with each other (pain avoidance vs. incorporation of healthy habits to manage pain). Pain is considered an obstacle in the acquisition of goals and pain catastrophizing reflects that conflict of goals in the tendency to avoid pain [33]. From this model, it is suggested that FM patients tend to meet objectives related to pain avoidance instead of doing physical activity. This condition is the result of the effect of pain catastrophizing, which could also influence the pain severity and long-term disability [34,35].

Despite the support received from motivational theories explaining the maintenance of pain and lack of adherence to walking patterns in FM patients, there are few studies on the influence of goals on symptoms such as pain or fatigue in chronic pain pathologies, as well as the role of pain catastrophizing in goal conflict and the deterioration of symptoms. Therefore, the main objective of the present study focuses on the exploration of the influence of a sedentary lifestyle on pain severity in terms of the preference for pain avoidance goals and the level of catastrophizing controlling the effect of pain chronicity. In summary, we hypothesize that the most sedentary FM patients tend to meet goals related to avoiding activities that enhance pain (Hypothesis 1; H1). In turn, patients who avoid more activities because of the pain are likely to perceive more intense pain after walking depending on the level of pain catastrophizing. That is, patients who display more avoidance behaviors will perceive more pain perception after activity if their thoughts about pain are more magnified (Hypothesis 2; H2). The results obtained in the proposed model may serve as a basis for future studies aimed at designing rehabilitative treatment programs, considering the conflict of personal goals as an influential variable in the health behaviors performed by patients with fibromyalgia.

## 2. Materials and Methods

### 2.1. Participants and Procedures

Participants were recruited voluntarily from different associations for fibromyalgia in the Community of Madrid and Castilla-La Mancha (Spain). All of them had previously met the diagnostic criteria for FM established by the American College of Rheumatology (ACR, 1990, 2010). Some of the patients were diagnosed with the 1990 ACR criteria: (a) chronic widespread pain of more than 3 months duration affecting at least 3 of the 4 quadrants of the body; and (b) pain or pressure from 11 of the 18 points chosen:–Occipital insertions of the suboccipital muscles;–Anterior cervical projection of the C5–C7 intertransverse spaces;–Midpoint of the upper border of the trapezius;–Origin of supraspinatus;–Second chondrosternal junction;–Two centimeters distally from the epicondyle;–Upper outer quadrant of the buttock;–Posterior aspect of the greater trochanter;–Adipose cushion of the inner aspect of the knee.

Other patients were diagnosed with the 2010 ACR criteria consisting of a Widespread Pain Index (WPI) and a Symptom Severity Scale including the following: fatigue, unrefreshing sleep, cognitive problems, and somatic symptoms (e.g., extreme tiredness, dry eyes, and blurred vision or irritable bowel syndrome) [36,37]. Despite their differences, previous literature has shown a good agreement between the 1990 and 2010 ACR criteria for FM diagnosis [38]. In addition to the diagnosis of fibromyalgia (ACR, 1990, 2010), other inclusion criteria to participate in the present study were the following: women being more than 18 years of age, providing written consent to participate and having a medical prescription of walking, but no present physical impairments for any physical activity. Those women who were less than 18 years of age, had other comorbid chronic pain problems (e.g., specific, or nonspecific chronic low back pain or chronic fatigue syndrome) or severe psychiatric diagnoses, were excluded as study participants. A large percentage (80–95%) of the patients diagnosed with FM are women, which is why we considered it appropriate to include only women in this study, to ensure the homogeneity of the sample.

The study was approved by the Bioethics Committee at Rey Juan Carlos University (Reference PI17/00858). All procedures performed in this study involving human participants were in accordance with the ethical standards of the institutional research committee and with the 1964 Helsinki Declaration. All participants signed informed consent forms and gave permission to use their evaluation data for research purposes. To preserve the privacy of our participants, each patient was assigned a numerical code to prevent recognition by the research team.

This study is part of a larger project consisting of three phases. Specifically, this study is part of the third phase that focuses on the study of pain catastrophizing and the conflict goals related to pain in walking behavior. Therefore, a design based on an observational analytical laboratory study is proposed.

First, a preliminary evaluation was carried out to obtain information on the demographic and clinical characteristics such as the type of medication administered, or the time elapsed since the fibromyalgia diagnosis. Self-report questionnaires (i.e., International Physical Activity Questionnaire, Pain Catastrophizing Scale, and Goal Pursuit Questionnaire) were also administered to evaluate the variables of interest. Subsequently, participants were asked to perform the Six-Minute Walk Test (6MWT). The purpose of this test is to measure the maximum distance a person can cover during a period of 6 min walking as fast as possible. The instruction given to each individual participant is to walk as fast as possible and as comfortably as possible for the stipulated time. The 6MWT is a test frequently used for fibromyalgia patients and recommended by the Spanish Society of Rheumatology for Fibromyalgia [39,40]. At the end of the walking test, an evaluation was carried out to determine the pain intensity.

Initially, 90 women with fibromyalgia were contacted, of whom 76 completed the evaluation protocol and performed the walking test.

### 2.2. Instruments

#### 2.2.1. Prior to Carrying Out the 6-Minute Walking Test

**Sociodemographic and clinical data**. First, open-ended questions were asked with the intention of collecting information from the participants, such as age, educational and employment level, marital status, time elapsed since the diagnosis of FM (years), and medication prescribed. All of these data were collected to describe the sociodemographic and clinical characteristics of the sample.

**Preliminary sedentary behavior**. The short form of the International Physical Activity Questionnaire (IPAQ-s) is known to be a valid and reliable self-reporting instrument for measuring walking behavior and types of physical activity as a function of intensity levels (intense or moderate) in clinical populations (adults aged 15–69 years) [41]. The questionnaire includes 7 items that identify the frequency (times per week) and duration (hours and minutes per day) of physical activity in the last week. To adapt the questionnaire to our aim and our study population, we use the last item that measures the time spent sitting as an indicator of sedentary behavior. This item measures the frequency (hours and minutes) of sitting for the last week and does not need accelerometers [41].

**Pain catastrophizing**. The Spanish adaptation of the Pain Catastrophizing Scale (PCS) was used for evaluating the negative cognitive and affective response to pain or expected pain [42]. This self-report instrument has three subscales: rumination, magnification, and helplessness. The PCS has 13-item self-report that measure the degree that they experience various pain-related thoughts or feelings. Scores range from 0 to 52 with higher scores indicating stronger pain catastrophizing. The PCS has shown good criterion-related validity and excellent internal consistency [43]. In this study, we use the pain catastrophizing total scale with a Cronbach’s alpha of 0.93.

**Preference for pain-avoidance goals related to physical activity**. The Goal Pursuit Questionnaire-physical activity (GPQ-PA) (self-report instrument) was used [44]. The GPQ-PA is an adaptation of the Spanish version [19] of the Goal Pursuit Questionnaire (GPQ) [45]. The Goal Pursuit Questionnaire (GPQ) [35,45] measures the hedonic or achievement goals, which can be activated at the same time in one situation for people with pain. Based on the GPQ, and with the aim of assessing the conflict of goals in relation to physical activity, the GPQ-PA contains 5 items that assess the preference for a hedonic goal or a physical activity goal. Those items refer to different physical activities that vary in intensity (walking, fast walking, light physical activity, moderate physical activity, and intense physical activity). Participants must imagine the situation presented in a vignette and rate their agreement with a specific thought, which refers to their goals-preference in this specific situation (e.g., You are moving from one place to another walking—walking the dog, going to work or shopping. Doing so causes you more and more pain throughout your body. You are expected to make that journey today). The GPQ-PA presents adequate psychometric properties [44]. Scores range from 5 to 30 and higher scores indicate stronger preferences for a hedonic goal (avoiding pain) relative to achievement goals (maintaining the activity). In the present study, the Cronbach’s alpha was 0.85.

#### 2.2.2. Six-Minute Walk Test (6MWT)

The 6MWT is the test that has been chosen to measure the distance, previously marked on the ground, that a person travels back and forth for 6 min. The researchers who applied the test and collected the data were CEG and PCM. In studies on FM [39], the results of the 6MWT were significantly related to the FM impact of the FIQ-R self-report. Subsequently, the reliability of the test for assessing possible functional limitations of patients with FM or chronic pain has been confirmed [46,47,48].

#### 2.2.3. After Carrying Out the 6-Minute Walking Test

**Pain perception.** Pain perception refers to the painful messages that emerge in the Peripheral Nervous System (PNS) and are transmitted to the Central Nervous System (CNS) where they are interpreted. In relation to the previous literature and the aims of the present study, pain perception is related to catastrophizing beliefs, i.e., the personal meaning attached to the painful experience [30]. In addition, perception is also related to the anticipation of anticipated situations that may be painful and, therefore, to the avoidance of the activity [30,31]. Therefore, we posited the assessment of pain perception and not the assessment of sensitivity to painful stimuli. Pain perception was evaluated after performing the 6MWT by an item of the Brief Pain Inventory (BPI) [49]. This item ranges from 0 to 10, with higher scores reflecting higher pain. Pain perception scores below 4 are commonly regarded as “mild” pain, while scores above 7 are “severe” [50]. This instrument has good psychometric properties [51].

**Covariates.** The number of years since diagnosis (ad-hoc questionnaire) and mean pain perception during the previous week [49] were evaluated as covariates.

Both the pre- and post-6MWT instruments were administered by the entire research team (LGH, CEG, PCM, and CPP). In the case of the self-report questionnaires (IPAQ-s, PCS, and GPQ-PA), the researchers were present to clarify any possible doubts that might have arisen when the patients completed them and to ensure that no missing values remained. Similarly, the monitoring and data collection of the 6MWT was carried out by the same researchers. Subsequently, a member of the research team oversaw transferring the data to the statistical bases and LGH and CPP were in charge of analyzing the data collected from the study. All the personnel involved in carrying out this study had sufficient teaching and research training to collect and analyze the data provided by the patients.

### 2.3. Statistical Analysis

All data were analyzed in IBM SPSS Statistic 27 for Windows (IBM Corporation, Amonk, NY, USA). Simple descriptive statistics were planned for sample demographics, along with bivariate Pearson correlations to explore associations between study variables (previous sedentary behavior, preference for pain-avoidance goals related to physical activity, pain perception, and pain catastrophizing). Compared to the approach by Baron and Kenny [52], the mediation and moderated-mediation models were tested by a bias-corrected bootstrapped procedure in PROCESS macro to control Type I error rates [53]. According to Hayes’ criteria [54], significant Pearson correlations allowed us to pose our mediated–moderately mediated model. This type of statistical model consists of several regressions exploring the predictive effects of a sedentary lifestyle and the avoidance goal preference. Based on this, we employed the t-statistic to test the significance of the hypotheses posed on mediation and moderation. To confirm significance, in addition to t, we used the *p*-value. The *p*-values that were lower than 0.05, together with t, confirm the hypothesis.

We used two predefined models to test our hypotheses. The simple mediation model of the relationship between previous sedentary behavior and pain perception after the 6MWT, mediated by the preference for pain-avoidance goals (in relation to physical activity) controlling for the effect of previous general pain and time elapsed since diagnosis as a measure of chronicity (covariates) (H1) was tested with model 4 in PROCESS [55] (see Figure 1).

To explore the moderating effects of pain catastrophizing, on the relationship between sedentary behavior, preference for pain-avoidance goals (in relation to physical activity), and pain severity after the 6MWT (H2), considering the aforementioned covariates, we proposed a moderated-mediation model with model 14 (see Figure 2). The number of bootstrap samples was 10,000 and all confidence intervals were 95%.

## 3. Results

### 3.1. Descriptive Analyses and Correlations between Study Variables of the Proposed Moderated-Mediation Model

The mean age of participants was 55.05 (standard deviation = 7.70). More than half had secondary studies (51.3%). The remaining participants reported having completed primary studies (35.5%), having had university studies (7.9%), or not having studied (5.3%). Most of the women were married (81.6%). Nine percent of the participants were separated or divorced, 5.3% were single, and 3.9% were widowed. Twenty-two percent of women were housewives or were working. Nineteen percent were on sick leave, 11.8% were retired, and 14.5% of participants were retired specifically due to disability. Nine percent were unemployed. Participants had had an FM diagnosis for an average of 12.32 years (standard deviation = 9.27; range 1–48 years) and the mean pain score during the last week was 5.67 (SD = 2.40; range 0–10). Regarding the medication they took on a daily basis, 16 women took antidepressants (21%), 22 women took anti-inflammatory drugs (24.3%), and 38 women took other types of medication to regulate their glucose levels, blood pressure or allergic problems (54.7%). Regarding the physical characteristics of the sample, the mean value of the BMI (body mass index) was 28.48 kg/m^2^ (SD = 5.71), the mean weight was 72.03 kg (SD = 15.46), and the mean height was 158.91 cm (SD = 5.29). Table 1 shows the means, standard deviations, and Pearson correlations between the study variables. We found significant associations between all variables.

### 3.2. Testing the Mediation Effect of the Preference for Pain-Avoidance Goals Related to Physical Activity between Previous Sedentary Behavior and Pain Perception

As presented in Table 2, there was a significant mediating effect of the previous sedentary behavior on pain perception via a preference for pain-avoidance goals related to physical activity. The bias-corrected bootstrap confidence interval for the indirect effect did not cross zero, this supports Hypothesis 1. There was a significant effect of the previous sedentary behavior on the preference for pain-avoidance goals (path a) (a = 0.40; *p* = 0.004; [CI = 0.007–0.015]). The effect of a sedentary lifestyle on pain perception through avoidance activity preference was also significant (path b) (b = 0.55; *p* = 0.0047; [CI = 0.057–1.102]). The total effect of the previous sedentary behavior on pain perception was significant (path c) (c = 0.120; *p* = 0.038; [CI = 0.015–0.038]). The direct effect (path c’) was significant and the simple mediation model accounts for 20.16% of the variance in pain perception after the 6MWT (R^2^ = 0.2016; c’ = 0.90; [CI = 0.017–0.035]). In relation to the covariates, pain perception over the past week had an influence on the intensity of pain perceived by the patients after performing the walking-test (β = 0.77; *p* = 0.000; [CI = 0.61–0.93]). However, the time elapsed after receiving the diagnosis was not significant in the model (β = −0.003; *p* = 0.982; [CI = −0.072–0.026]). In summary, the results indicate that previous sedentary behavior promoted a preference for avoidance of pain-related activities. Furthermore, activity avoidance was associated with more pain after the 6MWT.

### 3.3. Testing the Moderated-Mediation Model Based on the Pain Catastrophizing Levels

Table 3 presents the results of the moderated-mediation analyses based on pain catastrophizing (high, medium, or low levels). The results indicated that previous sedentary behavior influenced pain perception (path c) (c = 0.90; *p* = 0.047; [CI = 0.017–0.035]) according to the preference for pain-related activity avoidance goals (path a) (a = 0.40; *p* = 0.004; [CI = 0.007–0.015]). The effect of the preference for avoidance goals on pain perception after the 6MWT was also significant (path b1) (b1 = 0.58; *p* = 0.021; [CI = 0.090–0.110]). In addition, the pain catastrophizing of FM patients moderated the effect of the preference for pain avoidance goals on pain perception after the 6MWT (path b3) (b3 = 0.027; *p* = 0.008; [CI = 0.401–0.950]). The results of the conditional indirect effects showed that the observed positive link was only significant if FM patients reported high levels of pain catastrophizing (path b2) (b2 = 0.57; *p* = 0.021; [CI = 0.09–0.107]). The preferences for pain-avoidance goals were not associated with pain perception if patients reported medium or low levels of pain catastrophizing (medium level effect = 0.001 CI 95% [−0.002–0.008]; low level effect = 0.002 CI 95% [−0.004–0.007]). Similarly, the previous general pain reported by patients only influenced the effect of the preference for pain avoidance goals on perceived pain after the walking-test (β = 0.770; *p* = 0.000; [CI = 0.601–0.934]). No significant effect of general pain was found in preferences for pain-avoidance goals (β = 0.062; *p* = 0.220; [CI = −0.038–0.164]). The time elapsed since diagnosis had no effect on the proposed model (β = −0.026; *p* = 0.848; [CI = −0.029–0.024]). In summary, more sedentary women reported greater pain after the 6MWT due to preference for pain-related activity avoidance and significantly moderated higher levels of pain catastrophizing (R2 = 0.314; c’ = 0.88; [CI = 0.019–0.032]) (see Table 4 and Figure 3).

## 4. Discussion

Much of the research maintains that there is a greater tendency to a sedentary lifestyle in women with FM compared to healthy women [18]. The main reason why FM patients are less active is their pain management and fear of movement [25]. Pain catastrophizing is positioned as a contributor to explain the perception of pain and, consequently, the level of activity in patients [33,34]. However, few studies have examined the relationship between sedentary behavior and pain severity after activity from a conceptual framework based on motivational theories of pain, which emphasize the conflict of goals caused by health behaviors vs. pain management. The contextual variables of greatest interest in previous research are pain catastrophizing and goal preference within a complex model, in which mood may have an influence [56]. Specifically, it has been postulated that more catastrophic individuals prefer hedonic goals aimed at avoiding the threat of pain, rather than performing exercise [57]. Based on this confirmation, it is proposed that long-term achievement goals as well as short-term hedonic goals may be associated with increased pain and disability. Based on this evidence, the present study examined the moderated role of pain catastrophizing and the mediated role of preference activity goals on the relationship between sedentary behavior and pain experienced after a laboratory walking test. Our results reveal that sedentary behavior correlated significantly with pain and the preference for pain-avoidance goals. This result reinforces the mediation of pain-avoidance goals that was studied in previous research linking walking behavior and pain or fatigue in FM [58,59]. Despite focusing our interest on the effect of the main variables of the model on pain perception, we consider that it is a previous step to know how the interpretation of the painful experience influences pain modulation at a biological level. Furthermore, our model explains the relationship between the preference for avoidance goals promoted by an unhealthy behavior (sedentary lifestyle). In this sense, these findings may serve as a useful tool to rehabilitative programs that promote activity as one of the elementary components in multimodal programs to improve the functionality of patients with chronic pain.

In accord with the literature, physical activity, sedentary behavior, and physical fitness are well-known determinants of health in FM [20,51]. As in a previous study, women who were highly sedentary regulated pain less effectively than patients who spent less time in sedentary behaviors [51]. On the other hand, animal studies that have established neurobiological relationships with sedentary behavior confirm that a sedentary lifestyle facilitates a decrease in the uptake of insulin-like growth factor (IGF-I), which is necessary for the promotion of physical exercise. Consequently, the lack of uptake of this hormone weakens the functionality of the hippocampus and, therefore, promotes the subsequent development of neurodegenerative diseases [60]. The hypothalamus is a brain region that is connected to the amygdala, the insula, and the orbitofrontal cortex, forming the limbic system, in charge of processing the emotional response that people have when carrying out any behavior. Considering these relationships, it seems clear that the deterioration of the hippocampus due to a sedentary lifestyle can interfere with the decrease in the activity of the rest of the brain areas involved in the affective response and, therefore, alter the perception of pain due to the associated negative emotions and cognitions of the experience of pain [31,32,60]. Based on these findings, our results point to the influence of previous sedentary behavior on pain levels reported after the laboratory walking test. Specifically, sedentary behavior promotes higher levels of pain reporting after testing.

However, as is well known, fibromyalgia is characterized by widespread pain throughout the body [1,2]. Thus, we also raised the possibility that the pretest pain may have interfered with pain perception after walking for 6 min at a fast pace. Surprisingly, the effect of the pretest pain was only significant in the relationship between activity avoidance goals and post−6MWT pain, with the effect of the pretest pain on sedentary behavior not being significant. This finding may suggest to us that the chronic nature of this disease exerts an effect on the perceived pain levels after exertion. Thus, those patients who report high levels of previous pain would also report high levels of pain after walking because of the avoidance of physical exercise and the prevalence of catastrophic beliefs about the interference of pain with routine life. At the neurobiological level, pain catastrophizing influences attentional focus on potentially painful stimuli. Thus, individuals who may tend to catastrophizing have greater difficulty in diverting their attention and attribute a greater threat or harm to stimuli that are not painful, such as walking as an adapted physical exercise for fibromyalgia [60,61]. There is scientific evidence that pain catastrophizing leads to elevated affective pain scores and is related to different activation in brain areas such as the insular cortex, medial frontal cortex, and cerebellum, which are involved in the anticipation of painful stimuli or the avoidance of painful stimuli [62,63].

On the other hand, the model was confirmed regardless of the years of chronicity of the diagnosis. According to previous literature, patients with fibromyalgia often suffer the core symptoms of fibromyalgia, such as pain or fatigue, for many years despite not receiving a specific diagnosis by professionals at the time of symptom onset. This fact makes the symptomatology management and coping strategies generally maladaptive on the part of the women who do not receive supervision at the appropriate time (i.e., not engaging in physical activity to avoid pain intensity) [64]. In summary, despite the late diagnosis of FM, a highly relevant aspect in a multidisciplinary approach is the correct information for patients about their diagnosis, prognosis, and therapeutic possibilities. Within these possibilities, we emphasize muscle strengthening and toning exercises, as well as adapted aerobic exercise (e.g., walking), to improve not only pain perception, but also pain sensitization. We encourage future studies to attend to psychological factors such as pain catastrophizing or the preference for activity avoidance goals as major inhibitors of activity and potential risk factors in rehabilitation programs. However, further elaboration of the suggestions offered by our results is required in future research.

From the models of fear of movement, most of the research has been carried out taking adherence to physical activity as the variable under study. However, from these same models, explanatory studies on the influence of sedentary lifestyle are scarcer. In this context, the achievement of certain goals related to physical activity is associated with increased pain and fatigue in chronic pain patients [33,65]. Following the affective-motivational model of Karsdorp and Vlaeyen [45], in patients with chronic pain, there is a preference for short-term hedonic goals (avoidance of activity), despite being associated with an increase in symptoms. According to the associations raised by Pastor-Mira et al. [35] on activity patterns with a goal preference, our study considers the effect of regular sedentary behavior, stating that the most sedentary participants will suffer more pain and fatigue after the 6MWT. Additionally, the confirmation of hypothesis 1 indicates that goals related to pain-avoidance influence the increase in pain in those women who are more sedentary. We know that pain is one of the major inhibitors of the physical activity of women with FM [66,67] and given that patients do not undertake physical activity because they are avoiding a greater intensity of pain, implies that a lack of adherence to this pattern, so necessary in its multidisciplinary treatment. The present study suggests adherence to physical activity as an evidence-based approach with the intention of minimizing the negative consequences on the health patients.

Fear of pain is an influential variable in the preference for short-term hedonic goals. According to previous contributions, fear and avoidance is the result of a behavioral pattern that is not associated with the disease, and that leads to an exaggerated perception of pain [33,68]. In accordance with this approach, a catastrophic perception of pain leads to an excessive fear of pain, which implies an increase in pain and disability [33]. Pain-avoidance goals were positively associated with pain catastrophizing [35]. This result, together with our results on the significant relationship with pain severity after the 6MWT, support the findings offered by motivational models of pain-avoidance. Studies based on these models consider that catastrophic beliefs about pain and a preference for pain-avoidance goals predict increased pain and fatigue as the two main symptoms of FM [35,69]. Based on these findings, we proposed pain catastrophizing as a variable that acts as a moderator for the preference for hedonic goals and perceived pain after the walking test. Our second hypothesis helped us to confirm that previous sedentary behavior influenced later pain levels due to the preference for avoidance goals in those women who presented high excessive misinterpretations about pain. Also, the effect of pain catastrophizing on the activation of certain areas of the brain involved in attention to pain, pain-associated emotion, and motor behavior (physical activity vs. sedentary lifestyle), means that it may be beneficial to accompany physical and psychological therapy with other types of techniques or activities that modify the threat value that patients place on certain stimuli. Therefore, interventions based on the alteration of attention, or the modification of a perceived threat may be beneficial for those patients with FM who tend to present catastrophic beliefs about their symptomatology.

### Limitations and Strengths

The present study has some limitations that should be considered. First, it should also be acknowledged that there was a relatively low sample size assessed in this study and, in addition, the sample is made up of women only, which prevents generalization of the results. However, this study’s sample is similar to the sample sizes of previous studies based on the 6MWT in patients with chronic pain [48,70]. The nature of the present study, an observational analytical laboratory study, allows us to offer an approximation of the relationships between the variables of interest from the motivational approach to the behavior, but it prevents the generalization of long-term results. To alleviate this limitation, future studies should focus on exploring these relationships from the approach of longitudinal designs. In relation to the proposed model, sociodemographic (age, marital status, and education) or clinical (time of diagnosis) variables have not been considered as possible covariates interfering with the moderated-mediation model. It is suggested that the effect of these factors could be controlled for in future studies. Second, all instruments used are self-report measures. In relation to IPAQ, cut-off points for physical activity and sedentary behavior may also create measurement errors in FM patients due to these results being affected by the type of activities performed and the setting of the study. For this reason, the use of accelerometers is recommended as an objective measure that favors the correct interpretation of the results offered by this self-reporting in FM patients [71]. However, according to the previous literature, the use of accelerometers is not necessary to interpret sedentary behavior in this population [72,73].

Despite these limitations, the results of the current study increase our understanding of the relationship between pain catastrophizing and the preference of different goals in FM from a motivational perspective about pain. One of the strengths of this study is the use of the 6MWT, which has been commonly used in FM patients as an objective measure of functional capacity [69,70]. On the other hand, this study may help to explore the conflict goals related to pain (in physical activity situations) as one of the main reasons for the pain experienced after physical activity in FM women. Based on these results, motivational interventions can be of great help to improve the adherence to physical activity as one of the main components of multidisciplinary fibromyalgia treatment. Finally, our moderated-mediation model provides a good starting point in focusing on the contextual variables (conflict goals) related to the catastrophic perceptions about pain, responsible for the increase in pain in women who start an activity from a sedentary disposition.

## 5. Conclusions

In summary, a multidisciplinary approach based on attention to psychological factors such as catastrophic beliefs about pain and internal conflict about avoidance vs. activity goals could help patients with FM to overcome their medical condition and cope with the difficulties associated with the disease. Since walking is an accessible, easy exercise with favorable health outcomes for patients, enhancing the intention to initiate this behavior as a rehabilitation guideline within the multidisciplinary context is encouraged. To this end, by exploring the catastrophism towards pain and fear of movement, which can act as a promoter of sedentary behavior, the intensity of these beliefs can be reduced so that the adherence to walking behavior might be increased.

## Figures and Tables

**Figure 1 biomedicines-11-00154-f001:**
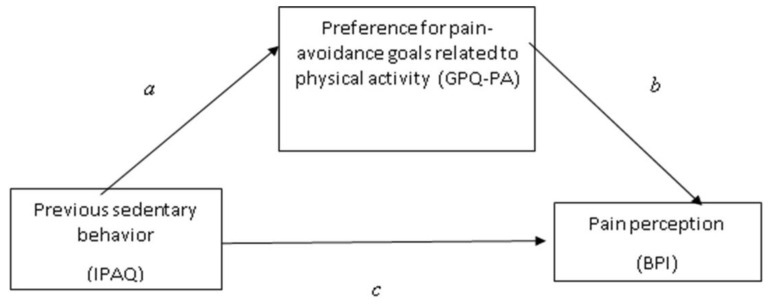
A conceptual and statistical diagram of a simple mediation model.

**Figure 2 biomedicines-11-00154-f002:**
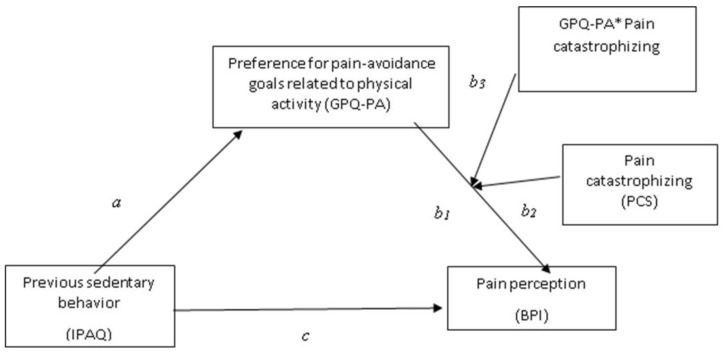
A conceptual and statistical diagram of a moderated-mediation model.

**Figure 3 biomedicines-11-00154-f003:**
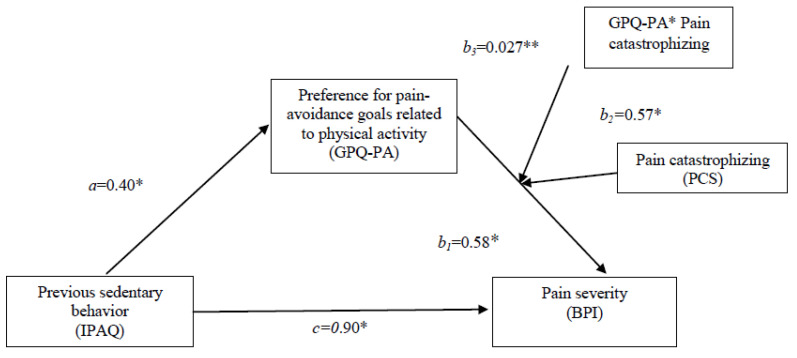
The moderated-mediation model showing the standardized analysis of previous sedentary behavior on pain through the preference for pain-avoidance goals and moderated by pain catastrophizing (model 14). Note: * *p* < 0.05; ** *p* < 0.01.

**Table 1 biomedicines-11-00154-t001:** Bivariate Pearson correlations between the observed variables (*n* = 76).

Variables	Theoric Range	M	SD	1	2	3	4
1. Sedentary behavior (minutes)	-	250.06	21.28	--			
2. Pain perception (a.u.)	0–10	6.01	2.54	0.10 **	--		
3. Pain catastrophizing (a.u.)	0–52	21.26	12.05	0.10 **	0.32 **	--	
4. Preference for pain-avoidance goals (a.u.)	0–30	4.71	1.05	0.84 *	0.23 **	0.27 **	--

Note: ** *p* < 0.01; * *p* < 0.05.

**Table 2 biomedicines-11-00154-t002:** The mediation model showing the effect of previous sedentary behavior on pain after the 6MWT via a preference for pain-avoidance goals.

Effect (Path)	Coeff	SE	t	*p*	LLCI	ULCI
Previous sedentary behavior → Pain-avoidance goals *(a)*	0.40	0.06	2.73	0.004	0.007	0.015
Pain-avoidance goals→ Pain perception *(b)*	0.55	0.28	2.01	0.004	0.057	1.102
Direct effect of previous sedentary behavior→ Pain perception *(c’)*	0.90	0.013	1.72	0.047	0.017	0.035
Total effect of previous sedentary behavior→ Pain perception *(c)*	0.120	0.013	1.87	0.038	0.015	0.038
**Bootstrap Results for the Indirect Effect**						
	**Effect**	**Boot SE**	**Boot LLCI**		**Boot ULCI**	
Indirect effect of previous sedentary behavior→ Pain perception	0.20	0.03	0.003		0.010	

Note. Coeff.: coefficient; SE: standard error; LLCI: lower level of the 95% confidence interval; ULCI: upper level of the 95% confidence interval.

**Table 3 biomedicines-11-00154-t003:** The moderated-mediation model showed the effect of pain catastrophizing on pain perception via the preference for pain-avoidance goals.

Outcome →	Pain Perception						Preferences for Pain-Avoidance Goals					
Predictor	Coeff	SE	t	*p*	LLCI	ULCI	Coeff	SE	t	*p*	LLCI	ULCI
Previous sedentary behavior (*c’*)	0.900	0.006	0.540	0.047	0.017	0.035	0.400	0.06	0.730	0.004	0.007	0.015
Preference for pain-avoidance goals (*b*_1_)	0.580	0.007	0.726	0.021	0.090	0.110						
Pain catastrophizing (*b*_2_)	0.570	0.024	2.371	0.021	0.090	0.107						
Previous sedentary behavior × Pain catastrophizing (*b*_3_)	0.027	0.27	0.140	0.008	0.401	0.950						
	R^2^ = 0.314 *F*(2.675) = 4.000, *p* = 0.038						R^2^ = 0.084 *F*(0.527) = 1.116, *p* = 0.472					

Note. Coeff.: coefficient; SE: standard error; LLCI: lower level of the 95% confidence interval; ULCI: upper level of the 95% confidence interval.

**Table 4 biomedicines-11-00154-t004:** The conditional effects of pain catastrophizing levels on pain perception via the preference for pain-avoidance goals.

Pain Catastrophizing	Scores	Effect	Boot SE	Boot LLCI	Boot ULCI
Low	−12.05	−0.260	0.003	−0.004	0.007
Medium	0.000	0.001	0.003	−0.002	0.008
High	12.05 *	0.570	0.021	0.09	0.107

Note. * Significant effect; SE standard error; LLCI lower level of the 95% confidence interval; ULCI upper level of the 95% confidence interval.

## Data Availability

The data presented in this study are available on request from the corresponding author. The data are not publicly available due to privacy.

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
