# Peer review of "Sedentary Behavior and Pain after Physical Activity in Women with Fibromyalgia—The Influence of Pain-Avoidance Goals and Catastrophizing"

_biomedicines, 2023, doi:10.3390/biomedicines11010154_

Round 1
Reviewer 1 Report
Title: Sedentary Behavior and Pain after Physical Activity in Women with Fibromyalgia. The Influence of Pain-Avoidance Goals and Catastrophizing
This article seems well built and brings evidence of a phenomenon not yet fully understood and that certainly deserves further study.
Some points of revision are provided below
Abstract
- I suggest to expand the abstract ……..including the results (data with p values)
Introduction
- Unclear the rationale of this study please to expand the introduction section….to better justify this study
Materials and Methods
- The authors need to include the anthropometric data (body mass/height) to better replicate this study and for future meta-analysis
- Unclear if the patient signed the informed consent as standard guidelines
Line 106 …..the authors wrote (Questionnaire is known to be valid and reliable instrument to measure walking behavior and types of physical activity as a function of intensity levels) but this sentence should be justified with an appropriate reference
Unclear how the 6-Minutes Walking Test was performed …..please check and justify with an appropiate reference
Table 1. please to include the u.m. (as a.u.)
Author Response
Reviewer 1:
Title: Sedentary Behavior and Pain after Physical Activity in Women with Fibromyalgia. The Influence of Pain-Avoidance Goals and Catastrophizing
This article seems well built and brings evidence of a phenomenon not yet fully understood and that certainly deserves further study.
Some points of revision are provided below
Abstract
- I suggest to expand the abstract ……..including the results (data with p values)
Response: Thank you for your appreciation. Based on the comments made, we have added the significant results found in the moderate mediation model based on high levels of pain catastrophizing. On the other hand, in accordance with the suggestions proposed by the editor, we have proceeded to reduce the number of words in the abstract to 200 words maximum. In the abstract we contemplate the most relevant findings of our study as emphasized by the reviewer, also following the editor's suggestion.
Introduction
- Unclear the rationale of this study please to expand the introduction section….to better justify this study
Response: we support the comment made by the reviewer. Therefore, we have expanded the information regarding the definition, prevalence and comorbidities of fibromyalgia. On the other hand, we have also expanded information on the efficacy of multimodal treatments in FM, emphasizing the need to promote physical exercise in patients to improve their quality of life and, specifically, the perception of pain. On the other hand, we have provided information on the neural mechanisms involved in the relationship between pain catastrophizing and pain perception. Finally, with the intention of clarifying our rationale, we have tried to express our interest in being the basis for further studies devoted to the design of rehabilitation programs that take into consideration our study variables.
Materials and Methods
- The authors need to include the anthropometric data (body mass/height) to better replicate this study and for future meta-analysis
Response: we have included the information suggested by the reviewer in the descriptive results section of the sample. (lines 319-320).
- Unclear if the patient signed the informed consent as standard guidelines
Response: we understand the importance of providing information on the ethical procedures carried out prior to the research. In accordance with the suggestion made, we have provided information on informed consent in the section on participants. (lines 167-173).
Line 106 …..the authors wrote (Questionnaire is known to be valid and reliable instrument to measure walking behavior and types of physical activity as a function of intensity levels) but this sentence should be justified with an appropriate reference
Response: we have added Ainsworth's reference [41] to explain the psychometric characteristics of the questionnaire in accordance with previous literature on its usefulness.
Unclear how the 6-Minutes Walking Test was performed …..please check and justify with an appropiate reference
Response: Reviewing the manuscript, we understand the difficulties of understanding regarding the explanation of the 6MWT. Therefore, we have provided information on the instructions indicated for initiating the test in fibromyalgia patients. Additionally, we have added references to previous literature that has used this test to objectively evaluate the activity of these patients (lines 235-241).
Table 1. please to include the u.m. (as a.u.)
Response: thank you for the appropriateness of your comment. In Table 1 we have included the units of measurement for each variable. One of them (sedentary behavior) is measured in minutes (objective unit) and the others are measured in arbitrary units. We have incorporated maximums and minimums in the variables measured with arbitrary units.

Reviewer 2 Report
In order to improve the manuscript, the following modifications should be included:
INTRODUCTION
1. The authors talk about "with a multitude of personal costs (…)", they should add more information on this point. Likewise, they should add more symptoms and signs of fibromyalgia, and recent epidemiological data on this syndrome, such as its incidence/prevalence, subjects affected, age at diagnosis, time of evolution, etc. All this information is relevant so that readers have a context of fibromyalgia syndrome. Include this new information in the final version of the manuscript.
2. Authors indicate "However, focusing therapeutic efficacy on reducing pain severity is an insufficient goal" The authors should clarify this sentence more and better, and include this information in the final version of the manuscript. If the authors included the pharmacological treatments used to relieve pain in fibromyalgia, and highlighted the ineffectiveness of these treatments, perhaps the relevance of physical exercise would be more justified. On the other hand, are combined therapies of physical exercise and pharmacological treatment effective? Please include all this information in the final version of the manuscript.
3. Authors indicate “However, the link of pain catastrophizing and pain is more complex because other contextual variables significantly influence this relationship”. What are these contextual variables that influence and that the authors indicate in the previous sentence? Include this information in the final version of the manuscript.
4. The objectives and hypotheses presented in the work are very important. However, the authors do not explain what neurobiological relationship exists between the catastrophizing of pain and the sensation and perception of greater pain, what neurobiological changes does the catastrophizing of pain trigger on the sensation and perception of pain? Authors must include this relevant information in the final version of the manuscript.
5. "A priori", the information that the authors try to find with the hypotheses raised in the manuscript, in which it can improve the quality of life of patients with fibromyalgia? How can the therapeutic strategies applicable to these patients be improved to improve their chronic pain and with it their quality of life? Please answer these questions and include this relevant information in the final version of the manuscript.
6. What is the novelty of this work compared to previous similar works? Include this information in the final version of the manuscript.
MATERIALS AND METHODS
7. Authors indicate “Participants were recruited voluntarily from different associations on fibromyalgia in Community of Madrid and Castilla-La Mancha (Spain). All of them previously met the diagnostic criteria for FM established by the American College of Rheumatology (ACR). Some of the patients were diagnosed with the 1990 ACR criteria. However, other patients received a later diagnosis, so they were diagnosed with the 2010 ACR criteria [20-21]”. A non-expert reader is unaware of the criteria that the authors speak of in the previous sentence. Please specify the 1990 and 2010 ACR criteria that have been used to diagnose a person with fibromyalgia, and specifically which of these criteria have been used specifically with the patients recruited in this study. It is understood that several doctors from different health centers have made this diagnosis, therefore, what is the degree of subjectivity and heterogeneity that these doctors have had when interpreting the ACR criteria and including the explored subject as a patient with fibromyalgia? Please clarify these points and include all this information in the final version of the manuscript.
8. Authors indicate “Other inclusion criteria to participate in the present study were the following: women being more than 18 years of age, providing written consent to participate and had a medical prescription of walking but not present physical impairments to carry out any physical activity. Those women who were less than 18 years of age, had other comorbid chronic pain problems (e.g., specific, or nonspecific chronic low back pain, chronic fatigue syndrome) or severe psychiatric diagnoses were excluded as study participants. The study was approved by the Bioethics Committee (omitted for blind review)”. Did the patients who were included with these new criteria also meet the 1990 or 2010 ACR diagnostic criteria for fibromyalgia? How many of the study participants included with the criteria described in the previous sentence met the ACR diagnostic criteria for fibromyalgia? Why have more participants been added with the previously indicated criteria? Lastly, authors must include detailed information on which Ethics Committees have approved the clinical procedures used in this paper, indicating the name of the Ethics Committee, the institution to which it belongs, the number of the approval file of the procedure or procedures used in this study, as well as the date of approval of said documents by the corresponding ethics committee. Please include all this information in the final version of the manuscript.
9. Why have only women been included? Fibromyalgia does not also affect men? The authors should clarify this point and include this information in the final version of the manuscript.
10. The authors describe a set of instruments that are applied to the patients recruited in the study. All of these instruments are questionnaires, who have applied these questionnaires to patients? Do the people who have applied these questionnaires have the same training criteria for their application? What degree of variability can there be when applying these questionnaires to recruited patients? Please clarify these points and include this information in the final version of the manuscript.
11. The submitted manuscript indicates that these instruments are applied before and after the so-called "The 6-Minutes Walking Test". The authors must describe in detail what this test consists of, indicate who has applied this test to the recruited patients, and how long after applying this exercise test the two instruments indicated in the manuscript have been administered to the patients. All this information is relevant, and should be included in the final version of the manuscript.
12. Why has the severity of pain been assessed only with a psychometric test? Why hasn't it also been evaluated with a quantitative test such as the von Frey filaments test that assesses mechanical allodynia, a characteristic that patients with fibromyalgia present? Please clarify these points and include this information in the final version of the manuscript.
RESULTS
13. In the first paragraph of the results, the authors describe a set of characteristics (e.g., educational level, work they do, drugs they take, etc.) of the recruited patients. However, the methodology does not indicate that these parameters have also been collected and analyzed from the recruited patients. Are all these sociocultural and therapeutic parameters indicated in the first paragraph relevant to the study presented? Do all these parameters influence the observed results? What is the degree of influence of these other parameters analyzed on the results of the questionnaires administered to the patients recruited in the study? Please include a section in the methodology where all the sociocultural parameters analyzed in the patients included in this study are specified, specifying their relevance to the study. Also, please clarify the previous questions and include all this information in the final version of the manuscript.
14. The authors should better explain Table 1, especially the Pearson correlations between the variables analyzed. This table is the summary of the relevant results of the present study, but it has nothing to do with these other sociodemographic variables. The authors should highlight this point, as it can lead to confusion for non-expert readers. Include this information in the final version of the manuscript.
15. It is very curious that the authors in the statistical analysis section do not talk about which statistical tests they apply to the results, however, in Table 1 they indicate that they perform a Pearson correlation analysis; in Table 2 they indicate that they perform confidence intervals with a "t" parameter that they do not describe, it is the Student's t test?, and the same happens with the rest of the tables presented in the results section. The conceptual models are very good, but they must be explained in detail, if they do not create too much confusion for the readers. Please clarify all these points and add all the information in the final version of the manuscript.
DISCUSSION
16. What are the motivational theories of pain that the authors talk about in the discussion? What are these theories? What is the neurobiological or psychobiological basis of these theories? Please include more detailed information in the final version of the manuscript.
17. The authors indicate “Given this gap, the present study examined the moderated role of pain catastrophizing and the mediated role of preference activity goals on the relationship between sedentary behavior and pain experienced after a laboratory walking test. Our results reveal that sedentary behavior correlated significantly with pain and preference pain-avoidance goals. This result reinforces the mediation of pain-avoidance goals that was studied on previous research linking walking behavior and pain or fatigue in FM [33-34” Why is this study and the parameters evaluated in it relevant in the area of psychology and/or psychobiology of pain? How can the results derived from this study improve the quality of life of patients with fibromyalgia? Please clarify all these points and include this relevant information in the final version of the manuscript.
18. The authors indicate “In line with literature, physical activity, sedentary behavior, and physical fitness are well-known determinants of health in FM [10,29]. As in a previous study, women who were highly sedentary regulated pain worse than patients who spent less time in sedentary behaviors [29]. Based on these findings, our results point to the influence of previous sedentary behavior on pain levels reported after the laboratory walking test. Specifically, sedentary behavior promotes higher levels of pain reporting after testing”. What is the neurobiological/psychobiological rationale that demonstrates that sedentary behavior promotes higher levels of pain after motor activity testing? Why does having a sedentary motor pattern mean that when doing physical activity it means a greater perception of pain that the subject feels? What areas of the nervous system could be involved in this perception of pain in a sedentary subject? The authors should discuss these aspects previously asked in greater depth. These are very important aspects, in which the authors must propose mechanisms involved. Please include all this relevant information in the final version of the manuscript.
19. The authors indicate “This finding may suggest to us that the chronic nature of this disease exerts an effect on perceived pain levels after exertion. Thus, those patients who report high levels of previous pain would also report high levels of pain after walking because of the avoidance of physical exercise and the prevalence of catastrophic beliefs about the interference of pain with routine life”. This sentence is a very relevant and important result that should be discussed in greater depth. What psychobiological mechanisms are involved in these relevant results? What plastic changes occur in the nervous system of fibromyalgia patients that lead to the results observed in this study? How catastrophic beliefs interfere in the perception of pain? What is the neurobiological/psychobiological substrate of this catastrophism? What plastic changes occur in the nervous system that lead to this catastrophizing in subjects with chronic pain, such as fibromyalgia patients? Please, it will be greatly appreciated if the authors answer all these questions, and include this relevant information in the discussion section of the manuscript.
20. The paragraph between lines 287-295, is a constructive criticism made by the authors of the delay in diagnosing fibromyalgia, and that this influences the perception of pain and the interference of catastrophizing on that perception of pain. What do the authors propose to improve the quality of life of patients who have been diagnosed late with fibromyalgia to improve the parameters analyzed in this study? Please include this relevant information in the final version of the manuscript.
21. It is very interesting what the authors discuss between lines 296-328. From what is commented in this part of the discussion, catastrophizing could be related to the model of fear of movement that leads to avoidance behaviors. Is it enough to carry out exposure therapies to physical activity and controlled exercise to change these avoidance behaviors associated with the catastrophizing of pain in patients with fibromyalgia? Or is it necessary to carry out therapies that lead to extinction of fear of movement that favor the generation of new neural circuits that lead to an improvement in the perception of pain associated with fibromyalgia? The authors had to discuss all these points in depth in the final version of the manuscript.
Author Response
Reviewer 2:
INTRODUCTION
- The authors talk about "with a multitude of personal costs (…)", they should add more information on this point. Likewise, they should add more symptoms and signs of fibromyalgia, and recent epidemiological data on this syndrome, such as its incidence/prevalence, subjects affected, age at diagnosis, time of evolution, etc. All this information is relevant so that readers have a context of fibromyalgia syndrome. Include this new information in the final version of the manuscript.
Response: Thank you for your suggestion and consideration in order to improve the understanding of fibromyalgia. In response to your suggestion, we have added information regarding the defining characteristics of the diagnosis of fibromyalgia, as well as the most recurrent conditions of patients (lines 36-42).
- Authors indicate "However, focusing therapeutic efficacy on reducing pain severity is an insufficient goal" The authors should clarify this sentence more and better, and include this information in the final version of the manuscript. If the authors included the pharmacological treatments used to relieve pain in fibromyalgia, and highlighted the ineffectiveness of these treatments, perhaps the relevance of physical exercise would be more justified. On the other hand, are combined therapies of physical exercise and pharmacological treatment effective? Please include all this information in the final version of the manuscript.
Response: Thank you for taking the time to appreciate the need to complete the information regarding the efficacy of treatments used in fibromyalgia. We have decided to complete the meaning of the sentence pointed out by the reviewer based on previous research on the efficacy of short-term pharmacological therapies. This is one of the most important arguments for opting for multidisciplinary treatments that include other techniques such as adapted physical exercise, with the intention of maintaining long-term benefits. This information can be seen reflected in the 55-70 lines.
- Authors indicate “However, the link of pain catastrophizing and pain is more complex because other contextual variables significantly influence this relationship”. What are these contextual variables that influence and that the authors indicate in the previous sentence? Include this information in the final version of the manuscript.
Response: we have completed the sentence by considering the motivational model proposed by Crombez [33-35] on goal conflict as the main contextual variable that interferes with pain management.
- The objectives and hypotheses presented in the work are very important. However, the authors do not explain what neurobiological relationship exists between the catastrophizing of pain and the sensation and perception of greater pain, what neurobiological changes does the catastrophizing of pain trigger on the sensation and perception of pain? Authors must include this relevant information in the final version of the manuscript.
Response: the chronicity of a pain diagnosis has its biological basis to explain also the role that cognitions play in the perception of painful information in people. We understand that this information is relevant in relation to the objectives pursued by the journal. For this reason, and following the reviewer's suggestions, we have provided brief information on the role of catastrophic beliefs in chronic pain processes in the Introduction section of the manuscript as well as the neural processes influenced by pain catastrophizing and self-reported pain perception (lines 89-104).
- "A priori", the information that the authors try to find with the hypotheses raised in the manuscript, in which it can improve the quality of life of patients with fibromyalgia? How can the therapeutic strategies applicable to these patients be improved to improve their chronic pain and with it their quality of life? Please answer these questions and include this relevant information in the final version of the manuscript.
Response: we have included a brief justification of our study after the explanation of the objectives and hypotheses (lines 130-133).
- What is the novelty of this work compared to previous similar works? Include this information in the final version of the manuscript.
Response: we have added a brief justification of the motivation for our study.
MATERIALS AND METHODS
- Authors indicate “Participants were recruited voluntarily from different associations on fibromyalgia in Community of Madrid and Castilla-La Mancha (Spain). All of them previously met the diagnostic criteria for FM established by the American College of Rheumatology (ACR). Some of the patients were diagnosed with the 1990 ACR criteria. However, other patients received a later diagnosis, so they were diagnosed with the 2010 ACR criteria [20-21]”. A non-expert reader is unaware of the criteria that the authors speak of in the previous sentence. Please specify the 1990 and 2010 ACR criteria that have been used to diagnose a person with fibromyalgia, and specifically which of these criteria have been used specifically with the patients recruited in this study. It is understood that several doctors from different health centers have made this diagnosis, therefore, what is the degree of subjectivity and heterogeneity that these doctors have had when interpreting the ACR criteria and including the explored subject as a patient with fibromyalgia? Please clarify these points and include all this information in the final version of the manuscript.
Response: We understand the doubt raised by the reviewer. The diagnostic criteria for fibromyalgia have been revised in recent years given the differences in the characteristics and demands of affected individuals in recent decades. In the case of our participants, our sample ranges in age from 18 to 80 years. Therefore, all were diagnosed according to the existing diagnostic criteria. The 1990 criteria include only pain as the main symptom, while the 2010 criteria include other types of symptoms such as fatigue, unrefreshing sleep, etc. Both diagnostic criteria maintain adequate concordance according to the study by Carrillo de la Peña et al. [38]. This explanation, along with the criteria, appears in the participants section (lines 138-157).
- Authors indicate “Other inclusion criteria to participate in the present study were the following: women being more than 18 years of age, providing written consent to participate and had a medical prescription of walking but not present physical impairments to carry out any physical activity. Those women who were less than 18 years of age, had other comorbid chronic pain problems (e.g., specific, or nonspecific chronic low back pain, chronic fatigue syndrome) or severe psychiatric diagnoses were excluded as study participants. The study was approved by the Bioethics Committee (omitted for blind review)”. Did the patients who were included with these new criteria also meet the 1990 or 2010 ACR diagnostic criteria for fibromyalgia? How many of the study participants included with the criteria described in the previous sentence met the ACR diagnostic criteria for fibromyalgia? Why have more participants been added with the previously indicated criteria? Lastly, authors must include detailed information on which Ethics Committees have approved the clinical procedures used in this paper, indicating the name of the Ethics Committee, the institution to which it belongs, the number of the approval file of the procedure or procedures used in this study, as well as the date of approval of said documents by the corresponding ethics committee. Please include all this information in the final version of the manuscript.
Response: the common inclusion criterion is that they meet the ACR diagnostic criteria (1990 or 2010). Therefore, the study participants met these criteria. However, we clarify this information in lines 155-156 of the manuscript. In addition, other inclusion criteria are requested, such as age and having walking behavior prescribed by their referring physician.
Regarding ethical issues, we have added the information requested by the reviewer (lines 167-170).
- Why have only women been included? Fibromyalgia does not also affect men? The authors should clarify this point and include this information in the final version of the manuscript.
Response: according to data on the prevalence of fibromyalgia, this health problem is more prevalent in women than in men. Given the ease of access to the participant sample, we considered it appropriate to choose women for our study. However, this may be considered an important methodological limitation due to the lack of generalizability of the results. Therefore, we have added this information in the procedure (lines 163-166) and in methodological limitations (lines 521-522).
- The authors describe a set of instruments that are applied to the patients recruited in the study. All of these instruments are questionnaires, who have applied these questionnaires to patients? Do the people who have applied these questionnaires have the same training criteria for their application? What degree of variability can there be when applying these questionnaires to recruited patients? Please clarify these points and include this information in the final version of the manuscript.
Response: this information has been added after the explanation of all the evaluation instruments (lines 257-266).
- The submitted manuscript indicates that these instruments are applied before and after the so-called "The 6-Minutes Walking Test". The authors must describe in detail what this test consists of, indicate who has applied this test to the recruited patients, and how long after applying this exercise test the two instruments indicated in the manuscript have been administered to the patients. All this information is relevant, and should be included in the final version of the manuscript.
Response: To provide more complete and adequate information about the 6MWT, we have prepared a new section containing information about the test and studies that support its reliability in clinical populations (lines 235-241).
- Why has the severity of pain been assessed only with a psychometric test? Why hasn't it also been evaluated with a quantitative test such as the von Frey filaments test that assesses mechanical allodynia, a characteristic that patients with fibromyalgia present? Please clarify these points and include this information in the final version of the manuscript.
Response: in accordance with our objectives and with the information previously provided in the Introduction, our approach focuses on the exploration of pain perception and its relationship with other variables of interest (sedentary lifestyle, avoidance goals and pain catastrophizing). For this reason, we consider it appropriate to administer self-reports that provide us with information on the personal meaning that patients give to the painful experience, instead of using assessment instruments on sensitivity to painful stimuli. To clarify these issues, we have chosen to modify the term "pain severity" to "pain perception" throughout the manuscript. We have also added clarifying information on the BPI assessment instrument (lines 243-254).
RESULTS
- In the first paragraph of the results, the authors describe a set of characteristics (e.g., educational level, work they do, drugs they take, etc.) of the recruited patients. However, the methodology does not indicate that these parameters have also been collected and analyzed from the recruited patients. Are all these sociocultural and therapeutic parameters indicated in the first paragraph relevant to the study presented? Do all these parameters influence the observed results? What is the degree of influence of these other parameters analyzed on the results of the questionnaires administered to the patients recruited in the study? Please include a section in the methodology where all the sociocultural parameters analyzed in the patients included in this study are specified, specifying their relevance to the study. Also, please clarify the previous questions and include all this information in the final version of the manuscript.
Response: thank you for your suggestion. We consider it appropriate to clarify the intentionality and the procedure that was carried out to collect this information. We have therefore added this information in the instruments administered prior to the 6MWT (lines 194-198).
- The authors should better explain Table 1, especially the Pearson correlations between the variables analyzed. This table is the summary of the relevant results of the present study, but it has nothing to do with these other sociodemographic variables. The authors should highlight this point, as it can lead to confusion for non-expert readers. Include this information in the final version of the manuscript.
Response: in accordance with the information provided in commentary nº 13, the sociodemographic and clinical data are only collected with the intention of defining the characteristics of the participants in our study. At no time are they considered as potential variables in the mediation-moderate model based on motivational theories (main objective of the study). Therefore, these correlations are not included in Table 1.
To make this clearer, we have modified the title of section 3.1 (lines 305-306). In addition, as suggested by the reviewer, we consider it appropriate to show as a limitation of the study the non-inclusion of clinical and sociodemographic variables as variables that interfere with the effects exerted by the variables of the proposed model (lines 528-531).
- It is very curious that the authors in the statistical analysis section do not talk about which statistical tests they apply to the results, however, in Table 1 they indicate that they perform a Pearson correlation analysis; in Table 2 they indicate that they perform confidence intervals with a "t" parameter that they do not describe, it is the Student's t test?, and the same happens with the rest of the tables presented in the results section. The conceptual models are very good, but they must be explained in detail, if they do not create too much confusion for the readers. Please clarify all these points and add all the information in the final version of the manuscript.
Response: after a careful reading of section 2.3, we understand the confusion we have caused the reviewer about the statistical tests performed. Mediation and moderation analyses are multiple regressions in which the predictor effects of certain variables are evaluated (in our case, the effect of sedentary lifestyle and preference for avoidance goals). Based on this, t does not refer to Student's t, but to the t coefficient that allows us to test the hypotheses that we set out in the model. In other words, the t-statistic allows us to test whether the regressions proposed in the mediation-moderate model are significant. We have decided to expand on this information in section 2.3. Statistical analysis to facilitate its understanding in the interpretation of the subsequent results.
DISCUSSION
- What are the motivational theories of pain that the authors talk about in the discussion? What are these theories? What is the neurobiological or psychobiological basis of these theories? Please include more detailed information in the final version of the manuscript.
Response: motivational theories are based on specifying the factors that influence people to decide on certain actions that drive behaviors. Based on this explanation, motivational models of chronic pain emphasize their interest in goal preference and other psychological variables such as mood and pain catastrophizing to determine why people tend to be sedentary and the effect it has on the severity of symptomatology. For a better understanding of these concepts, we have expanded the information in the first paragraph of the Discussion (lines 404-410).
- The authors indicate “Given this gap, the present study examined the moderated role of pain catastrophizing and the mediated role of preference activity goals on the relationship between sedentary behavior and pain experienced after a laboratory walking test. Our results reveal that sedentary behavior correlated significantly with pain and preference pain-avoidance goals. This result reinforces the mediation of pain-avoidance goals that was studied on previous research linking walking behavior and pain or fatigue in FM [33-34” Why is this study and the parameters evaluated in it relevant in the area of psychology and/or psychobiology of pain? How can the results derived from this study improve the quality of life of patients with fibromyalgia? Please clarify all these points and include this relevant information in the final version of the manuscript.
Response: we have added a brief justification of the usefulness of our model in future studies devoted to the design of rehabilitation programs at the end of the first paragraph of the Discussion.
- The authors indicate “In line with literature, physical activity, sedentary behavior, and physical fitness are well-known determinants of health in FM [10,29]. As in a previous study, women who were highly sedentary regulated pain worse than patients who spent less time in sedentary behaviors [29]. Based on these findings, our results point to the influence of previous sedentary behavior on pain levels reported after the laboratory walking test. Specifically, sedentary behavior promotes higher levels of pain reporting after testing”. What is the neurobiological/psychobiological rationale that demonstrates that sedentary behavior promotes higher levels of pain after motor activity testing? Why does having a sedentary motor pattern mean that when doing physical activity it means a greater perception of pain that the subject feels? What areas of the nervous system could be involved in this perception of pain in a sedentary subject? The authors should discuss these aspects previously asked in greater depth. These are very important aspects, in which the authors must propose mechanisms involved. Please include all this relevant information in the final version of the manuscript.
Response: In relation to what was explained in the Introduction on the neural mechanisms that relate pain catastrophizing to the perception of pain, we have provided information on the deterioration of certain brain areas involved in the emotional response to the painful experience and sedentary lifestyle to discuss the support of our model to previous studies developed under psychological and biological theories on pain (lines 428-440).
- The authors indicate “This finding may suggest to us that the chronic nature of this disease exerts an effect on perceived pain levels after exertion. Thus, those patients who report high levels of previous pain would also report high levels of pain after walking because of the avoidance of physical exercise and the prevalence of catastrophic beliefs about the interference of pain with routine life”. This sentence is a very relevant and important result that should be discussed in greater depth. What psychobiological mechanisms are involved in these relevant results? What plastic changes occur in the nervous system of fibromyalgia patients that lead to the results observed in this study? How catastrophic beliefs interfere in the perception of pain? What is the neurobiological/psychobiological substrate of this catastrophism? What plastic changes occur in the nervous system that lead to this catastrophizing in subjects with chronic pain, such as fibromyalgia patients? Please, it will be greatly appreciated if the authors answer all these questions, and include this relevant information in the discussion section of the manuscript.
Response: we have provided information on the influence of pain catastrophizing on attentional biases and how this process affects emotional responses associated with pain and reinforces pain anticipation/avoidance since, pain avoidance, can be understood to be sedentary in accordance with the objectives set out in this study. Additionally, we have emphasized the altered activity of the insula, hypothalamus, and cerebellum as structures directly involved in activity avoidance and promoted by catastrophic beliefs about pain. (lines 453-461).
- The paragraph between lines 287-295, is a constructive criticism made by the authors of the delay in diagnosing fibromyalgia, and that this influences the perception of pain and the interference of catastrophizing on that perception of pain. What do the authors propose to improve the quality of life of patients who have been diagnosed late with fibromyalgia to improve the parameters analyzed in this study? Please include this relevant information in the final version of the manuscript.
Response: we have provided a brief explanation of the potential of our findings for the development of future programs aimed at rehabilitating the physical functionality of patients (lines 468-475).
- It is very interesting what the authors discuss between lines 296-328. From what is commented in this part of the discussion, catastrophizing could be related to the model of fear of movement that leads to avoidance behaviors. Is it enough to carry out exposure therapies to physical activity and controlled exercise to change these avoidance behaviors associated with the catastrophizing of pain in patients with fibromyalgia? Or is it necessary to carry out therapies that lead to extinction of fear of movement that favor the generation of new neural circuits that lead to an improvement in the perception of pain associated with fibromyalgia? The authors had to discuss all these points in depth in the final version of the manuscript.
Response: according to the information provided on the neural mechanisms and the implication of pain catastrophizing on the attentional processes in pain, training in exercises that work on attention and the modification of the threatening meaning given to the symptom would be appropriate. This information is collected in lines 510-517.
-----
Reviewer 2:
The authors have made a great effort to respond to the questions and comments made by the reviewer, and have included most of the information in this second version of the manuscript. This has significantly improved the quality of this latest version. I believe that they have satisfactorily answered the questions raised by the reviewer, and that they have included relevant information in the manuscript. Therefore, I consider that this second version can now be published.
Response: Thank you for the positive assessment of our manuscript.

Round 2
Reviewer 1 Report
The authors improved a lot the article also if it's very hard to follow each point because the main document need to be revisited while the body mass and height the authors cannot write (Regarding the physical characteristics of the participating 317 sample, the body mass of the sample stood at 34 kg/m2.) this sentence is wrong....should include body mass and body height as mean / SD
Author Response
Response: thank you for your suggestion. First, we have reviewed the data concerning the body mass index, weight, and height of the participants. We have detected a flaw in the calculations, so the body mass index of the participant sample is 28.48 kg/m2. Additionally, and following the comments made by the reviewer, we have included the mean height (cm) and weight (kg) with their respective standard deviations (lines 320-321).
Reviewer 2 Report
The authors have made a great effort to respond to the questions and comments made by the reviewer, and have included most of the information in this second version of the manuscript. This has significantly improved the quality of this latest version. I believe that they have satisfactorily answered the questions raised by the reviewer, and that they have included relevant information in the manuscript. Therefore, I consider that this second version can now be published.
Author Response
Response: Thank you for the positive assessment of our manuscript.